# Updates in Cancer Cachexia: Clinical Management and Pharmacologic Interventions

**DOI:** 10.3390/cancers16091696

**Published:** 2024-04-27

**Authors:** Sudeep Pandey, Lauren Bradley, Egidio Del Fabbro

**Affiliations:** 1Department of Internal Medicine, Division of Hematology, Oncology and Palliative Care, Virginia Commonwealth University, Richmond, VA 23298, USA; sudeep.pandey@vcuhealth.org (S.P.); lauren.bradley@vcuhealth.org (L.B.); 2Department of Medicine, Division of Palliative Medicine, Medical College of Georgia, Augusta University, Augusta, GA 30912, USA

**Keywords:** cancer cachexia, cachexia, cancer, wasting syndrome

## Abstract

**Simple Summary:**

Cancer cachexia (CC) is a complex syndrome requiring a multimodal approach. Although a universally accepted definition and staging criteria for cancer cachexia remains elusive, there is general consensus regarding the importance of elements such as weight loss, muscle wasting, and poor appetite. Epidemic trends of obesity and overlapping conditions such as sarcopenia and frailty further complicate CC definition, staging, and relevant outcome measures. Despite progress in understanding the molecular mechanisms of CC, there is no single, consistently effective pharmacotherapy for CC and, unsurprisingly, there are variations among guidelines regarding management. Current pharmacologic research is focused on promising targeted treatments; however, a multimodal approach is likely to be more effective than any single therapeutic agent. This narrative review provides an update on non-pharmacologic and pharmacologic treatment and proposes a theoretical model for management of CC, which includes a multimodal therapeutic approach directed at the various mechanisms contributing to CC.

**Abstract:**

Despite a better understanding of the mechanisms causing cancer cachexia (CC) and development of promising pharmacologic and supportive care interventions, CC persists as an underdiagnosed and undertreated condition. CC contributes to fatigue, poor quality of life, functional impairment, increases treatment related toxicity, and reduces survival. The core elements of CC such as weight loss and poor appetite should be identified early. Currently, addressing contributing conditions (hypothyroidism, hypogonadism, and adrenal insufficiency), managing nutrition impact symptoms leading to decreased oral intake (nausea, constipation, dysgeusia, stomatitis, mucositis, pain, fatigue, depressed mood, or anxiety), and the addition of pharmacologic agents when appropriate (progesterone analog, corticosteroids, and olanzapine) is recommended. In Japan, the clinical practice has changed based on the availability of Anamorelin, a ghrelin receptor agonist that improved lean body mass, weight, and appetite-related quality of life (QoL) compared to a placebo, in phase III trials. Other promising therapeutic agents currently in trials include Espindolol, a non-selective β blocker and a monoclonal antibody to GDF-15. In the future, a single therapeutic agent or perhaps multiple medications targeting the various mechanisms of CC may prove to be an effective strategy. Ideally, these medications should be incorporated into a multimodal interdisciplinary approach that includes exercise and nutrition.

## 1. Introduction

“*You don’t necessarily have to get on the scales, you see the bones begin to protrude and feel the end is near*.” Patient [1]

Cachexia has long been recognized as a syndrome of wasting and progressive inanition. In the 21st century, CC remains a complex, underdiagnosed, and undertreated condition. Lack of a universally accepted definition, diagnostic criteria, and classification has impeded progress in both clinical trials and clinical practice throughout decades [2,3,4]. Efforts by professional organizations in providing management guidelines have raised awareness of the condition and improved clinical care; however, variation among guidelines persists. Our objective for this review is to provide updated information regarding clinical management and pharmacologic interventions for CC.

## 2. Definition and Staging

The consensus definition states that CC is a multifactorial syndrome characterized by an ongoing loss of skeletal muscle mass (with or without loss of fat mass) that cannot be fully reversed via conventional nutritional support and leads to progressive functional impairment. The pathophysiology is characterized by a negative protein and energy balance driven by a variable combination of reduced food intake and abnormal metabolism [5].

This definition pivots on involuntary weight loss (WL) of more than 5% over 6 months (or 2% when evidence of sarcopenia is present) and a body mass index (BMI) < 20 as prime diagnostic criteria. The evidence to support the use and significance of WL and BMI to diagnose CC is accumulating, even though the definition fails to address current trends toward excess weight and obesity [5,6,7,8]. (Figure 1) Recently, a grading system based on metabolic reserve and rate of weight loss by Martin et al. incorporated the two dimensions of %WL and BMI and correlated them with survival. The prediction of survival was independent of conventional prognostic factors including cancer site, stage, and performance status (PS). This study was large enough to validate the concept proposed within the international cachexia classification framework [9].

Oncology patients demonstrate a wide variability in body habitus, functional abilities, and biomarkers at the time of diagnosis and throughout anti-neoplastic treatment. To cater for this inter-patient variability, a framework to investigate and stage cachexia was proposed, classifying patients as no cachexia, pre-cachexia, cachexia, and refractory cachexia. Initial attempts to validate the stages based on specific criteria were unsuccessful; however, a cachexia staging score (CSS) that included %weight loss, a SARC-F (strength, assistance with walking, rising from a chair, climbing stairs—falls) questionnaire, ECOG PS (Eastern Cooperative Oncology Group Performance Status), appetite score, and abnormal biochemistry (white blood cell count, albumin, and hemoglobin) proposed by Zhou et al. [10], from a single center in China, was able to classify cachexia into the four stages, showing life expectancy was significantly different among the stages [7,11]. In addition, the CSS was validated by comparing differences in muscle mass, sarcopenia, symptom burden, and quality of life among the four groups. Subsequently, a study of 196 Japanese patients receiving palliative care reported ‘excellent prognostic discriminative power’ when using the CSS for the different stages of cachexia [11].

Another simple, objective, systemic-inflammation–based approach using C-reactive protein (CRP) and albumin (Glasgow Performance Scale—GPS) is a potential framework for identifying and treating CC. The GPS has a prognostic value independent of tumor stage and PS in a variety of advanced tumors and has evidence for effectively stratifying the treatment response to CC [8].

The requirement for a more dynamic model to stage cachexia is further supported by the current worldwide prevalence of overweight, obesity [12], and emerging research in sarcopenia. There is a paradoxical relationship between obesity and prognosis. Although obesity at the time of cancer diagnosis is generally associated with a better prognosis, sarcopenic obesity confers an adverse impact on survival [10,11]. Sarcopenia is characterized by a loss of skeletal muscle mass and function and predicts survival regardless of body weight. The European Working Group on Sarcopenia in Older People (EWGSOP) published a consensus paper, updating the clinical algorithm to diagnose and confirm sarcopenia and providing clear cut off points with the aim of increasing awareness of sarcopenia and its risk [13].

More research is required to develop a unifying classification system for cachexia that incorporates patient-reported outcomes, biomarkers, and body composition to facilitate staging, prognostication, and appropriate patient selection for clinical trials.

## 3. Clinical Impact

“*This bony thing shows up in the mirror every morning, and my eyes fall on this creature on the other side of the mirror*.” Patient [1]

The impact of cachexia in clinical practice ranges from increased mortality risk (up to 20% of cancer deaths) to negative body image (58.3% of the cancer patients reported a negative body image) [14,15] and family conflict [16]. Studies show cachexia shortens life [17], decreases response to treatment, increases the failure rate of complete treatments (patients with weight loss and non-small cell lung cancer (NSCLC) (*p* = 0.003) or mesothelioma (*p* = 0.05) more frequently did not to complete at least three cycles of chemotherapy) [18], worsens fatigue (stronger correlation between strength vs QoL (R > 0.33, *p* < 0.001) [19], contributes to lower QoL, and decreases PS [20,21,22,23]. Martin et al.’s survival model compared conventional covariates (cancer diagnosis, stage, age, and performance status) to a model which included BMI, weight loss, muscle index, and muscle attenuation (MA). Patients who experienced weight loss, sarcopenia, and low MA survived for only 8.4 months (95% CI, 6.5 to 10.3), regardless of their initial presentation BMI, compared to patients with none of these features who survived for 28.4 months (95% CI, 24.2 to 32.6; *p* < 0.001) [24]. 

Clinical research relating to CC is beset by there being only a few large randomized controlled trials (RCTs) and very low patient participation [25]. Low patient enrollment has been attributed to stringent inclusion and exclusion criteria, concerns regarding potential drug interactions with prescribed cancer treatment, fears surrounding randomization to placebo, and the demands of participating in a study while receiving treatment for a serious illness. There are also high dropout rates, particularly in patients with advanced cancer, due to non-study-related hospitalization, study non-adherence, and admission to hospice [25]. Increasing accrual in cachexia trials may require less stringent entry criteria, fewer burdensome outcome measures, and increasing awareness through patient education tools (for example via healthcare professionals, the internet, and apps). This may help engage and empower patients leading to increased recruitment and retention in clinical research [25,26,27].

## 4. Assessment Tools

“*I was five feet from him before he could figure out who it was. I cried, because he was a very, very good friend of mine. It seemed to confirm the fact that I was so skinny*.” Patient [1]

To ensure timely clinical and metabolic intervention, early detection of CC is integral. Delay could contribute to uncontrolled symptoms, poorer quality of life, and more rapid progression to the refractory stage of cachexia. At-risk patients should be routinely screened via a standardized procedure (Figure 2). Unfortunately, an early diagnosis may be hindered simply by a lack of awareness among healthcare professionals.

Questionnaires: Without a systematic inquiry, symptoms such as anorexia may not be identified since patients volunteer only a few symptoms relative to their total symptoms experience [28,29,30]. The Edmonton Symptom Assessment Scale (ESAS) has identified a high prevalence of multiple symptoms in ambulatory patients with cancer [31]. However, although the ESAS assesses symptom severity, it includes some (pain, nausea, depression, anxiety, and appetite) but not all nutrition impact symptoms (NISs) [32]. For a more comprehensive evaluation of NIS, an additional assessment measure such as the Patient-Generated Subjective Global Assessment (PG-SGA) can be included. PG-SGA is an American-Dietetic-Society-endorsed questionnaire that screens for additional reversible factors contributing to poor oral intake. A validated short-form version, the PG-SGA SF, can be completed in a few minutes and provides diagnostic and prognostic value for patients with cancer [33].

Physical performance tests show a correlation with important clinical outcomes including survival [11]. Tests such as hand grip strength and Short Physical Performance Battery (SPPB) are widely used in aging populations, and their use is growing in oncology [34]. To assess physical performance and screen for sarcopenia, hand grip strength testing using a handheld dynamometer, a simple, affordable tool [35,36,37], while SPPB is recommended to evaluate the severity of sarcopenia [38] and is predictive of clinical outcomes including mortality and healthcare utilization [39]. The SPPB is a composite test including an assessment of gait speed, balance, and a chair stand test. A SPPB score < 10 is predictive of decreased overall survival (OS) in older patients with leukemia [40], increased post-operative complications [41], and adverse events with lower chemo completion in NSCLC [42,43].

**Figure 2 cancers-16-01696-f002:**
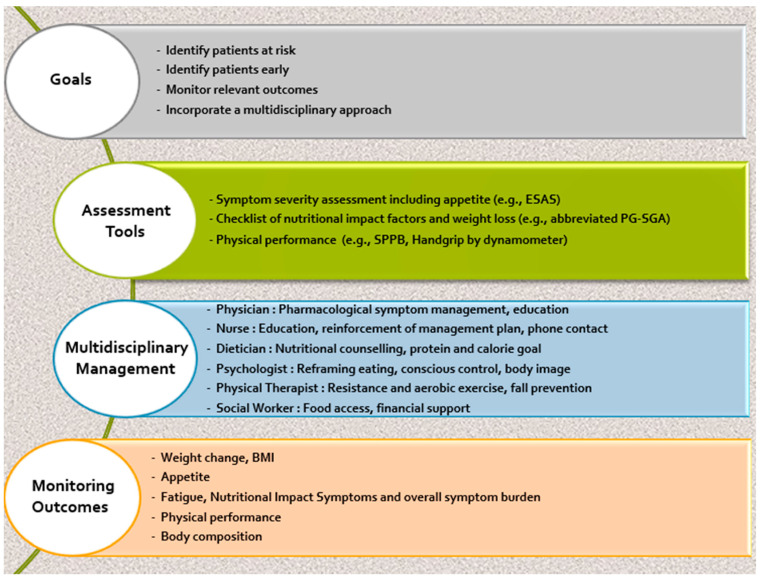
A framework for the management of patients with cancer cachexia in daily practice. (ESAS, Edmonton Symptom Assessment Scale; PG-SGA, Patient-Generated Subjective Global Assessment; SPPB, Short Physical Performance Battery) [44].

Body composition: Cancer patients who are overweight or obese may have a normal or elevated body mass index, despite reporting weight loss and experiencing profound muscle wasting. Sarcopenic obesity is strongly related to a worse prognosis [45] and higher risk of chemotherapy-related adverse effects [46]. Studies performed on pancreatic cancer patients revealed that both skeletal muscle and visceral adipose tissue (VAT) loss is associated with poor survival [47,48]. Body composition assessment via dual energy X-ray absorptiometry (DEXA scan) [49], Computed Tomography (CT) scans, or ultrasound (US) may identify patients with sarcopenia.

Though body composition assessment (fat and muscle) is not conducted regularly in clinical practice, CT imaging is a useful opportunistic tool given that CT is performed at the time of initial cancer staging and also later for assessment of tumor size and response. This can provide potentially relevant longitudinal information towards quantification of muscle area and attenuation [24,50].

Muscle ultrasound is gaining increasing attention for measuring muscle mass and diagnosing sarcopenia due to its safety, noninvasiveness, low cost, and real-time characteristics [51]. A recent systematic review and meta-analysis published [52] showed that muscle ultrasound has only low-to-moderate accuracy for diagnosing sarcopenia.

Bioimpedance (BIA) relies on the different electrical properties of fat and muscle and is a relatively easy to use and nonburdensome for patients. BIA is not as accurate as CT imaging or DEXA [53].

## 5. Mechanism of Cancer Cachexia

“*I cooked a lot, I baked a lot… He ate what I made because I made what he liked… We always ate together… He wouldn’t eat if I didn’t eat, so I stopped eating when he stopped eating*.” Caregiver [1]

The management of CC is challenging because the syndrome has multiple inter-related mechanisms and psychosocial domains (e.g., access to adequate nutrition). However, in the past decade, significant progress in basic science research has led to important advances that deepen our understanding of the complex interplay among various biological processes contributing to CC. Unfortunately, as yet, there are no FDA-approved agents for CC, underscoring the unmet need for effective pharmacologic therapy.

The primary causes of CC are thought to be dysregulation of proinflammatory and acute-phase proteins and neuroendocrine dysfunction due to cancer–host interactions (Figure 3). This leads to a variable combination of inflammatory and metabolic changes including systemic inflammation, proteolysis, lipolysis, lipid mobilization, increased resting energy expenditure, and a decrease in protein synthesis, lipogenesis, and appetite. At a molecular level, these changes can be reflected by an increase in TNF-α, CRP, IL-1, IL-6, GDF-15 levels, ghrelin resistance, and insulin resistance and a decrease in testosterone levels. The clinical impacts observed in cancer patients are anorexia, weight loss, decreased lean body mass, reduced overall survival, decreased QoL, and decreased physical activity [54].

## 6. Management of Cancer Cachexia

“*At first I thought we were in limbo, nobody cared, that we couldn’t turn to anybody… nobody seemed to help us… we just had to cope on our own…. I thought that someone should have come and spoke to us as a family to tell us what to expect… when he wasn’t feeling well for a doctor or what or who to turn to*.” Caregiver [1]

Because CC is a multidimensional clinical syndrome, linear strategies such as nutritional supplementation are likely to fail unless accompanied by a multimodal approach to management. Ideally, management should entail assessment of the patient’s clinical condition and nutritional status followed by an evaluation of appropriate, available treatment options. This should culminate in a personalized multimodal approach [55].

In this review, our theoretical model for future management of CC includes management strategies supported by clinical studies that include RCT’s and observational trials. Although most clinical trials have used single therapeutic agents as interventions, there are a growing number of multimodal studies that combine medication and non-pharmacologic therapy. Evidence from preclinical models for treatment of CC are generally avoided in this review.

## 7. Non-Pharmacologic Management

Exercise

Exercise is hypothesized to attenuate the effects of CC by modulating muscle metabolism, insulin sensitivity, hypogonadism, and systemic inflammation. Most of the evidence for improved outcomes is associated with aerobic exercise; however, resistance training is also effective and increasingly employed as an intervention. A meta-analysis of 34 resistance-training trials in patients with cancer or survivors showed a greater lean body mass (LBM) increase compared to controls (*p* = 0.004) [56]. Combinations of resistance and aerobic exercise also demonstrate capacity for improving outcomes such as muscle strength and functional ability [57].

A systematic review found exercise is safe, improves quality of life, and is beneficial for muscular and aerobic fitness both during and after treatment for cancer [58]. Similarly, a recent systematic review in patients with incurable cancer reported improvements in physical endurance and depression scores, despite limited studies [59]. Most trials also show improvements in Patient Reported Outcomes (PROs) for fatigue, an important component of the CC syndrome. However, an RCT comparing 8 weeks of supervised exercise training twice weekly with usual care in 231 patients with advanced cancer reported improved physical performance but no effect on fatigue scores [60].

The challenge remains that some patients, particularly those with advanced cancer, may stop participating in an exercise program with disease progression. Ideally, exercise programs should be tailored to each patient’s unique needs and then adjusted depending on their capabilities during and after anti-neoplastic treatment. Despite advanced disease, exercise appears to be a safe and practical approach for patients, even in the hospice setting [61,62].

Nutrition

The goal of nutritional support is to maintain adequate energy and nutrient intake but also enable a patient to enjoy eating and participate in meals with others as a component of social life [32]. Dietary counseling and a thorough assessment to identify contributors towards decreased intake are important, including the quality (e.g., macro and micronutrient content) and duration of nutritional support.

Evaluating an individual patient’s precise nutritional needs is challenging since cancer may increase resting energy expenditure (REE) (hypermetabolic), while others are unchanged (eumetabolic) or hypometabolic. In addition, despite an increase in REE, the total energy expenditure (TEE) may be unchanged because of a reduction in physical activity. Accurate measurement of nutritional needs using a metabolic chamber or even indirect calorimetry are ideal but not feasible in daily practice [63]. The estimated nutritional needs for oncology patients as recommended by guidelines includes goals of 25–30 kcal/kg/day and 1.2–1.5 g protein/kg/day [64]. Increased meal/snack frequency, and the use of oral liquid nutritional supplements or energy-dense foods, represent potentially effective therapies in combination with dietary counseling.

Three systematic reviews suggested that receiving oral nutritional supplements alone, without dietary counseling, were not effective, emphasizing the need for an interdisciplinary approach to CC [65,66,67]. A systematic review also identified potential flaws in trial design of nutritional support interventions that might contribute to negative outcomes in food intake, body weight (BW), and QoL [68,69].

Nutrition Impact Symptoms (NIS)

Pharmacologic treatment, using readily available inexpensive medications, is a cornerstone of managing symptoms that contribute to decreased nutritional intake. (Figure 4) NISs such as nausea, vomiting, early satiety, constipation, depression, anxiety, severe pain, mucositis, and dysgeusia can contribute to decreased caloric intake. A retrospective study of 151 patients with solid tumors referred to a specialized CC clinic found a median of three NISs and five or more NISs in 15% of patients. Early satiety was the most common reported symptom (62%) and responded well to metoclopramide (79%) [70]. Patients with advanced cancer often have gastroparesis and dysmotility; metoclopramide enables the stomach to accommodate food and improves motility [71,72].

Nausea and vomiting may be treatment-related (chemotherapy, radiation, and surgery) or chronic, non-treatment-related. For non-chemotherapy-induced nausea and vomiting (non-CINV) guidelines recommend metoclopramide as the therapy of choice rather than other medications associated with sedation (prochlorperazine, promethazine, and scopolamine) and constipation (ondansetron) [73]. Olanzapine is an attractive alternative for non-CINV, following a positive pilot trial [74] and phase III trials showing benefit for chemotherapy-induced nausea and vomiting (CINV) [75] and appetite (see below in pharmacologic treatment).

Constipation, exacerbated by medications such as opioids and ondansetron may contribute to early satiety, abdominal pain, and nausea. This can be effectively managed with laxatives such as polyethylene glycol and senna, although few published trials compare bowel regimens [76].

Depressed mood and anxiety may decrease appetite and can lead to malnutrition [77]. Depression should be managed appropriately with counseling and antidepressants, if indicated. Mirtazapine and olanzapine are useful agents for both depression and nausea [78].

The evidence supporting the use of mirtazapine for weight gain is mixed. A small, single-arm trial of mirtazapine in nondepressed patients with cachexia produced weight gain of 1 kg or greater in about one-quarter of participants within 4 weeks [79,80]. However, in a recent randomized trial, mirtazapine 15 mg at night did not improve appetite or body weight compared to a placebo in patients with cachexia [81].

Metabolic abnormalities such as hypogonadism (see below in androgens), thyroid dysfunction, and vitamin B12 and D deficiencies may contribute to fatigue, muscle weakness, and poor appetite [82]. Vitamins should be prescribed if their serum levels are low and testosterone replacement therapy should be considered in symptomatic patients with low serum testosterone levels [83].

There are no consistently effective therapies for dysgeusia; however, in a small RCT, Dronabinol enhanced chemosensory perception and improved the taste of food when compared with a placebo [84,85]. A trial of zinc sulfate may be considered [86,87] because zinc has few side effects in comparison to dronabinol, and an RCT showed improved taste alterations in patients with head and neck cancer compared to a placebo. Other causes of weight loss such as gastrointestinal obstruction should be identified, especially if they are reversible. For example, esophageal obstruction can be addressed by either stent placement or endoscopic dilation.

## 8. Pharmacologic Interventions

Although no single agent has shown exceptional benefit in treating or ameliorating CC, several phase II and phase III studies have reported improved clinical outcomes.

 Current Agents

Corticosteroids

Corticosteroids (CSs) include several agents with variable glucocorticoid, mineralocorticoid, and anti-inflammatory potency. Prednisolone, methylprednisolone, and dexamethasone are used most frequently and are typically prescribed to patients with very advanced disease [88,89].

Several RCTs investigating the effects of CSs show improve symptoms of anorexia and fatigue [90,91]. Two randomized placebo-controlled trials validate CS efficacy. In patients with advanced head and neck or gastrointestinal cancer, 4 mg dexamethasone twice daily for 14 days significantly improved fatigue and anorexia compared to a placebo [89]. The second study compared 16 mg methylprednisolone twice daily for 7 days to a placebo and similarly to the previous study showed improvements in fatigue and anorexia [92]. Despite short term improvements in appetite and fatigue, no studies have demonstrated any benefit on LBM or survival, and prolonged use of CSs increases the risk of complications such as infection and proximal myopathy. The optimal dose and duration of CSs are unclear, and limited data are available to recommend one CS over another. Dexamethasone is considered preferable due to its lower mineralocorticoid effect. Common toxicities include candidiasis, edema, cushingoid changes, depression, and anxiety [93]. 

Progestational agents/Progestins

Systematic reviews indicate megestrol acetate (MA) increases appetite and body weight compared to a placebo [94,95,96]. A Cochrane review in 2013 [94] evaluated 23 trials in patients with cancer, including 928 patients with gastrointestinal or pancreatic cancer. Approximately one in four patients taking MA for cachexia (e.g., to treat cancer or AIDS) had an increase in appetite, while one in twelve experienced weight gain. However, no consistent improvement in QoL was observed and no data on muscle mass or physical function were reported [94].

Dyspnea, edema, impotence, and thromboembolic phenomena were more common in patients taking MA and mortality increased with higher doses. Because the median treatment duration was 8 weeks and follow-up lengths were short, the authors suggest adverse events may be even more pronounced with prolonged use. In the analyzed trials, MA was used in doses of 160–800 mg/day and weight improvement appeared higher for doses > 160 mg/day.

MA-induced weight gain is predominantly fat or fluid, rather than muscle [97]. Although some earlier studies showed improvement in fatigue as a secondary endpoint [98,99], there are concerns that prolonged suppression of gonadal and adrenal function by MA could exacerbate symptoms such as fatigue and poor libido [100]. MA may also have an antianabolic effect and can decrease muscle size [101]. These questions concerning the adverse effects of MA remain and its use must be carefully considered and managed. If an improvement in appetite and a gain in fat mass are desirable goals for a patient, MA remains one of the most potent orexigenic agents available, but its use must be weighed against the potential for increased mortality risk, thromboembolism, and adrenal and androgen suppression.

Cannabinoids

Synthetic cannabinoids, Dronabinol and Nabilone are FDA approved for chemotherapy-related nausea in patients who do not respond to conventional antiemetics [102]. For AIDS patients who report anorexia, Dronabinol is an approved treatment modality [103]. However, in patients with CC, the benefits appear limited, and psychotropic side effects are a concern, particularly at higher doses. In an RCT, dronabinol improved the taste and protein consumption in oncology patients with dysgeusia but showed no improvement in body weight compared to a placebo [85].

A multicenter trial of 289 patients with advanced cancer compared the effects of cannabis extract (CE) (standardized for 2.5 mg THC and 1 mg cannabidiol), delta-9-tetrahydrocannabinol (2.5 mg twice daily), and a placebo on appetite and QoL [104]. Appetite improved in all three groups, but there were no significant differences between the groups for appetite, QoL, or cannabinoid-related toxicity. An earlier phase II study showed a higher risk of adverse psychotropic effects and dropouts in patients taking higher doses of dronabinol (5 mg vs. 2.5 mg) [105]. Combination therapy with MA also appears to have no additive benefit. An RCT compared MA and dronabinol combination therapy versus either agent alone for appetite stimulation in 469 patients with lung or gastrointestinal cancer. MA was superior to dronabinol alone, and combination therapy did not provide any additional benefits [106]. Similarly, although well tolerated, Nabilone did not significantly improve appetite or QoL [107] in a placebo-controlled RCT of 47 patients with advanced NSCLC.

Olanzapine

Olanzapine (OLZ) is an atypical, second-generation antipsychotic that acts on multiple receptors including those for adrenalin, dopamine, serotonin, histamine, and muscarine. In clinical use for psychiatric conditions, OLZ was found to cause more weight gain than other antipsychotic drugs [108]. In patients with cancer, OLZ was safe and effective for nausea and vomiting in two important RCTs: first, for the prevention of chemotherapy-induced nausea and vomiting (CINV) and, secondly, for treating non-CINV [109,110,111,112]. OLZ at 5 mg/day significantly reduced non-CINV in 30 patients with advanced cancer compared with a placebo [74].

Besides efficacy for CINV and non-CINV, OLZ is also accumulating evidence for improving appetite in patients with cancer. A single-arm, dose-escalation study of 39 patients with CC receiving anti-neoplastic treatment tested OLZ doses ranging from 2.5 mg to 20 mg daily. OLZ only had a modest effect in altering the trajectory of weight loss [113]. Another RCT involving 80 patients with advanced gastrointestinal or lung cancer compared MA alone with a combination of MA and olanzapine for 8 weeks. The combination arm yielded significant improvements in appetite and BW [114].

ASCO recently published a CC guideline update stating, ‘for adults with advanced cancer, clinicians may offer low-dose olanzapine once daily to improve weight gain and appetite’ [110]. The recommendation was prompted by an RCT of 124 patients with locally advanced or metastatic gastric, hepatopancreatic biliary, and lung cancers receiving OLZ 2.5 mg once a day for 12 weeks or a placebo, along with chemotherapy (all patients received OLZ 5 mg daily × 4 days for CINV). The OLZ arm had a greater proportion of patients with weight gain of >5% (60% vs. 9%, *p* < 0.001) and an improvement in appetite on visual analog scale (43% vs. 13%, *p* < 0.001). Patients on OLZ also experienced better QoL, nutritional status, and lesser chemotoxicity [115].

Non-steroidal anti-inflammatory drugs (NSAIDs)

NSAIDs block the cyclooxygenase pathways and inhibit prostaglandin production, that cause inflammation and pain. Based on the premise that inflammation is a main driver of cachexia, the number of anti-inflammatory drug trials in cachexia may increase, despite risks of gastrointestinal bleeding and kidney injury with NSAIDs. For select patients suffering from CC and pain, NSAIDs have the potential for dual benefit.

A systematic review published in 2012 of thirteen studies (six controlled trials and seven observational trials) found either improvement or stabilization of BW or LBM in eleven of thirteen trials with few reported side effects. Unfortunately, because small sample sizes and methodological flaws impaired the quality of the studies, NSAID use for CC outside clinical trials was not recommended [116].

Thalidomide

Proinflammatory cytokines including tumor necrosis factor α (TNF-α) probably play a prominent role in the pathogenesis of CC. Thalidomide is an inhibitor of TNF-α synthesis and is a potentially inexpensive, rational approach to treat CC.

A Cochrane review of thalidomide in CC included only three studies [117]. Patients with esophageal cancer showed no benefit and poor tolerability to thalidomide [118], despite earlier trials in pancreatic cancer [119] and esophageal cancer [120] reporting improvements in LBM and minimal side effects after 4 weeks of 200 mg/day. A phase II trial found improved appetite and minimal side effects with doses of 50 and 100 mg [121]. Other studies of thalidomide have experienced challenges with poor patient accrual and attrition [122]. More trials are warranted, especially with low-dose thalidomide, which is better tolerated. Currently, there is insufficient evidence for clinical practice to refute or support the use of Thalidomide.

Fish oil or Eicosapentanoic acid (EPA)

EPA, an n-3 fatty acid, has antitumor and anticachectic effects in the murine MAC-16 colon adenocarcinoma model [123]. EPA initially showed some benefits for cachexia, anorexia, and fatigue in patients with pancreatic cancer [124]. However, a subsequent RCT of 518 patients with advanced gastrointestinal or lung cancer found pure EPA at a dose of 2 g or 4 g daily for 8 weeks was no better than a placebo for survival, weight, or other nutritional variables [125].

Although two small RCTs in patients with NSCLC at initiation of first-line chemotherapy showed improved weight and muscle mass [126,127], these preliminary studies must be tempered by four systematic reviews (2007 to 2022) reporting insufficient evidence for EPA in the management of CC [128,129,130,131]. Although no serious adverse effects were reported, abdominal discomfort, belching, nausea, and diarrhea often affected QoL.

 New agents

Androgens

Hypogonadism is common in male patients with cancer and is associated with increased symptom burden including fatigue, anorexia, and diminished libido [96,97,98,99,132]. Low testosterone may be a result of chronic inflammation or secondary to medications such as opioids, corticosteroids, MA, or anti-neoplastic therapy (e.g., crizotinib). In some wasting disorders such as HIV, anabolic–androgenic steroids have evidence for improving muscle mass and strength [133]; however, in oncology patients the results are inconsistent.

A three-armed RCT randomized 496 patients with CC to dexamethasone 0.75 mg qid, megestrol acetate 800 mg orally every day, or fluoxymesterone 10 mg orally bid. Fluoxymesterone was significantly inferior to MA and dexamethasone for appetite improvement [134], while another RCT in NSCLC patients found nandrolone (200 mg weekly for 4 weeks) was no better than a placebo for improving BW [135].

Theoretically, selective androgen receptor modulators (SARMs) produce greater anabolic effects with fewer virilizing effects. A phase II RCT of Enobosarm for patients with CC reported increased LBM and physical function compared to baseline, with minimal side effects [136].

Phase III trials (POWER Trials) with Enobosarm in patients with NSCLC beginning first-line chemotherapy showed inconsistent results regarding lean body mass and stair climb power [137]. Despite an abstract publication several years ago reporting a clinically significant effect on muscle mass (but no consistent effects on muscle function), no publication has reported on the results of the POWER Trials and Enobosarm was not granted regulatory approval in the U.S. and Europe [138].

Although testosterone replacement therapy (TRT) for CC is not yet supported by evidence, there are preliminary trials demonstrating improvement in cachexia-related outcomes such as fatigue. A trial of TRT reported no improvement in fatigue scores after 4 weeks, but, by day 72, fatigue improved significantly with intramuscular testosterone compared to placebo injections, indicating greater benefit with prolonged use. A significant improvement in ECOG status was found after 4 weeks; however, these encouraging results are limited by the trial’s small sample of 29 participants [139]. A larger, adequately powered, multicenter, placebo-controlled trial is underway using TRT (topical gel) for cancer-related fatigue, body composition, and muscle function in men ≥ 55 years [140].

Beta Blockers

There is evidence for Beta blockers mitigating muscle wasting in conditions such as burns (placebo-controlled trial with propranolol) and in CC (observational studies with atenolol and propranolol [141,142]. Espindolol is a non-selective β blocker targeting three potential mechanisms relevant to CC: reducing catabolism (non-selective β blockade), reducing fatigue and thermogenesis (central 5-HT1a antagonism), and increased anabolism (partial β2 agonism) [143].

The ACT-ONE trial [144] randomized 87 patients with stages III/IV colorectal cancer or NSCLC and CC in a ratio 3:2:1 [high dose: 10 mg twice daily, placebo, and low dose: 2.5 mg twice daily]. Espindolol 10 mg BID significantly reversed weight loss, improved fat free mass, and maintained fat mass when given over 16 weeks. Hand grip strength also improved significantly and no adverse events were noted. A larger, multicenter Espindolol trial is underway for CC.

Ghrelin and ghrelin mimetics

Ghrelin, is a peptide hormone which functions as an endogenous ligand for the growth hormone (GH) receptor and exhibits a dose-dependent, GH-releasing activity [145]. In preclinical cachexia models, ghrelin has shown beneficial effects on appetite, food intake [146,147], lean body mass [148], gastrointestinal motility [149], energy metabolism, and proinflammatory cytokine expression [150]. After establishing safety and efficacy in otherwise healthy patients [151], a pilot study in seven patients with cancer showed 31% higher energy intake with IV ghrelin than with a placebo and no adverse effects [152].

A phase II RCT [153] in patients with esophageal cancer receiving cisplatin-based neoadjuvant chemotherapy reported increased food intake and higher appetite visual analog scale (VAS) scores in patients receiving ghrelin. Notably, fewer adverse events related to anorexia and CINV were reported in the ghrelin group compared to the control group. Ghrelin’s use for CC is hindered by the subcutaneous mode of administration and daily frequency. Furthermore, although studies have consistently confirmed safety, ghrelin has the potential for increasing insulin-like growth factor 1 [154] and, theoretically, promoting tumor progression.

Anamorelin, an oral ghrelin mimetic, is the first drug approved for CC, although this is limited to Japan since Anamorelin is not approved in North America or Europe. An integrated analysis of two phase II trials [155] using Anamorelin for CC found increased lean body mass over 12 weeks compared to a placebo. In larger, paired phase III trials (ROMANA 1 and 2) [156], Anamorelin significantly improved lean body mass, bodyweight, fat mass, and appetite compared to a placebo in patients with NSCLC. Unfortunately, the co-primary endpoint hand grip strength, a surrogate marker for physical function, did not improve significantly.

Anamorelin is approved in Japan for treating CC patients with NSCLC, gastric cancer, pancreatic cancer, and colorectal cancer. Recent post hoc analyses of Anamorelin in subgroups of Japanese patients with NSCLC confirmed the efficacy and tolerability of Anamorelin regardless of age and PS [142].

Myostatin and proinflammatory cytokine inhibitors

Myostatin, a member of the transforming growth factor (TGF) family, is an extracellular cytokine that negatively regulates skeletal muscle mass. Myostatin is upregulated in many conditions of muscle wasting [157,158] and is a potential therapeutic target [159], although outcomes of early phase clinical trials targeting the myostatin pathway are either heterogenous or not yet published.

Anti-inflammatory drugs with a broad spectrum of action such as NSAIDs and corticosteroids have demonstrated benefit but, unfortunately, also have off-target effects. Directly targeting specific proinflammatory cytokines involved in the pathogenesis of CC is preferable, in theory. Unfortunately, drugs targeting TNF-alpha such as etanercept and infliximab did not improve clinic outcomes such as LBM in patients with pancreatic cancer [160,161]. In a phase 1 study, adult patients with metastatic cancer were given escalating doses of MABp1 (monoclonal antibody against anti-interleukin-1α) monotherapy. MABp1 was well tolerated, and patients showed a median weight gain of 1 kg from baseline and no dose-limiting toxicities [162]. Targeting the myostatin pathway and neutralization of inflammatory cytokines is an exciting concept in treating CC but, for now, has very limited clinical research data.

Anti-Growth Differentiation Factor 15 (GDF-15)

Elevation in circulating GDF-15 is associated with cachexia and reduced survival in patients with cancer [163,164,165,166]. GDF-15, like myostatin, is a member of the TGF-β superfamily and has exceptional potential for serving as both a marker and a therapeutic target in managing CC.

In preclinical studies, GDF15-associated weight loss has been described [167,168,169,170]. A therapeutic monoclonal antibody, acting on the GDF-15 signaling pathway, was found to reverse excessive lipid oxidation in tumor-bearing mice, thereby preventing CC even under calorie-restricted conditions. Benefits included the prevention of excessive protein catabolism and inhibition of muscle wasting, leading to improved function [171]. In another preclinical study, GDF-15 neutralization improved muscle function and physical performance in a murine CC model [172].

These positive preclinical studies led to the development of Ponsegromab, a monoclonal antibody that binds and blocks GDF-15. In a phase 1 study, 10 patients with CC (NSCLC, colorectal, or pancreatic) and elevated serum concentrations of GDF-15 received open-label subcutaneous Ponsegromab every three weeks (Q3W) for twelve weeks. Ponsegromab was safe, well tolerated, and showed preliminary evidence of efficacy, including a mean weight gain of 6.5% at 12 weeks. A multicenter RCT, is currently underway, determining whether Ponsegromab is an effective strategy for weight gain and enhancing physical performance in CC.

## 9. Multimodal Therapy for the Cancer Cachexia

While novel single agents exhibit promising outcomes, a more effective approach may be simultaneous, multifaceted therapy targeting the different mechanisms contributing to CC [173]. (Figure 4) Several studies have used a combination of pharmacologic agents for CC. For instance, progestin given in combination with EPA, L-carnitine, and thalidomide significantly increased appetite, LBM (*p* = 0.007), and spontaneous physical activity, although there was no placebo arm for comparison [174].

Beta blockers [142] and insulin [175] in combination with nonsteroidal anti-inflammatories (NSAIDs) demonstrated benefits in reducing elevated resting energy expenditure, attenuated weight loss, and improved survival (*p* = 0.03).

Non-pharmacologic interventions such as exercise and nutrition should be considered in any multimodal intervention for CC. An RCT in 58 patients compared usual care with 12 weeks of exercise training (biweekly) combined with nutritional counselling three times per week. The treatment arm showed significantly increased protein intake and improved wellbeing (attributed to decreased nausea and vomiting) [176]. A phase III RCT of 328 patients with untreated metastatic esophagogastric cancer compared the standard of care (SC) against early interdisciplinary supportive care (ESC). The trial did not specifically enroll patients with CC; however, poor appetite and fatigue were the most severe reported symptoms. ESC was provided by a team of GI medical oncologists, oncology nurse specialists, dietitians, and psychologists; patients in the SC group received standard oncologic care alone. The ESC group resulted in improved overall survival [177].

The MENAC feasibility trial was the first of its kind to compare a combination therapy of NSAIDs (Celecoxib), nutritional counseling, oral nutritional supplements (ONAs) enriched in EPA, and physical exercise to standard treatment for 6 weeks in patient with lung and pancreatic cancer [62]. This combination therapy was found to be feasible and safe, with the results of a follow-up multimodal, multi-site, phase III trial expected soon. The intervention includes ibuprofen (1200 mg/day), omega-3 fatty acids (2 g EPA and 1 g docosahexaenoic acid), supplementation with 542 kcal and 30 g of protein, and a home-based exercise program consisting of resistance training three times/week in addition to aerobic training 2 times/week.

Ideally, combination therapy should ideally have additive or even synergistic effects. The current evidence suggests that even an effective single pharmacologic agent would work most effectively when combined with dietary counseling, nutrition supplementation, symptom management, and exercise.

## 10. Conclusions

Because the mechanisms of CC are multifactorial, linear strategies alone have not been successful in managing this complex syndrome. A comprehensive multimodal approach using pharmacologic and nonpharmacologic interventions seems the most effective way to reverse or stabilize weight loss and muscle wasting. The optimal strategy involves a customized treatment plan that identifies the principal mechanism(s) of an individual’s weight loss and integrates management into personal goals of care. New anti-cachexia/targeted agents show promise in preliminary clinical studies, but larger phase III RCT’s are required to establish their overall safety and efficacy. We now have several promising pharmacologic interventions, but any specific anti-cachexia intervention will require integration into a multimodality approach.

## Figures and Tables

**Figure 1 cancers-16-01696-f001:**
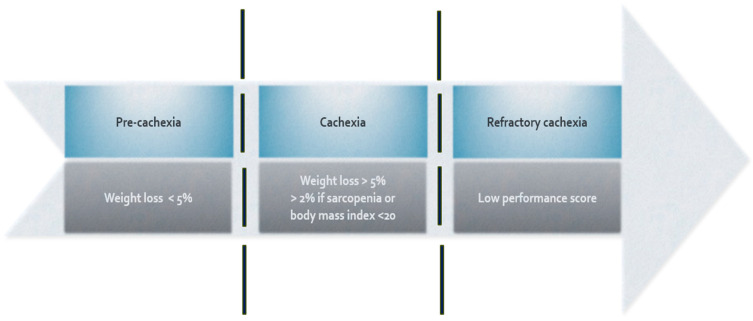
Proposed stages of cachexia [6].

**Figure 3 cancers-16-01696-f003:**
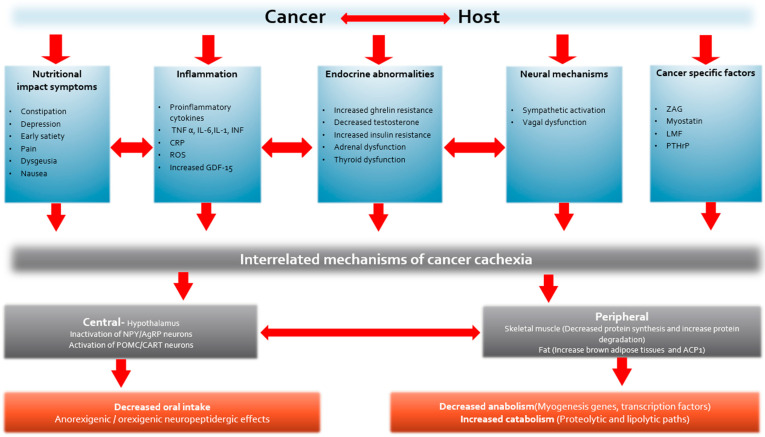
Simplified theoretical model demonstrating the complex interplay of different mechanisms producing cachexia. These mechanisms should not be viewed in isolation, since they are all inter-related, e.g., proinflammatory cytokines have a profound effect on the endocrine system, but there is also reciprocal modulation. TNF: Tumor Necrosis Factor; IL: Interleukin; INF: Interferon; CRP: C-Reactive Protein; ROS: Role of Oxidative Stress; GDF: Growth Differentiation Factor; ZAG: Zinc-α_2_-Glycoprotein; LMF: Lipid-Mobilizing Factor; PTHrP: Parathyroid Hormone-Related Protein; NPY: Neuropeptide Y; AgRP: Agouti-Related Protein; POMC: Pro-opiomelanocortin; CART: Cocaine and Amphetamine-Regulated Transcript; ACP: Acid Phosphatase.

**Figure 4 cancers-16-01696-f004:**
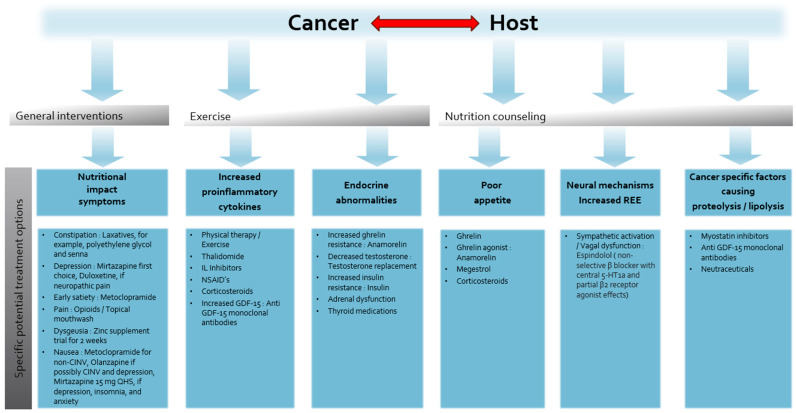
Simplified summary of multimodal management of cancer cachexia.

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
