# Peer review of "Updates in Cancer Cachexia: Clinical Management and Pharmacologic Interventions"

_cancers, 2024, doi:10.3390/cancers16091696_

Round 1

Reviewer 1 Report

Comments and Suggestions for Authors

-       The abstract summarizes the key points of the manuscript, including the persistent challenges associated with cancer cachexia, current management strategies, and future directions; however, it could benefit from elaborating on specific outcomes of interventions mentioned, such as improvements in quality of life or survival rates, to provide a more comprehensive overview;

-  The introduction contextualizes the significance of CC as a clinical syndrome, highlights existing challenges in its diagnosis and management, and sets the stage for subsequent discussions on potential solutions, providing a solid foundation for the reader to understand the scope and importance of the topic;

-    The reference to the European Working Group on Sarcopenia in Older People (EWGSOP) consensus paper underscores the importance of integrating emerging research findings into clinical practice;

-    Where possible, quantify the effects of CC on mortality, treatment response rates, quality of life scores, etc. This would provide a clearer understanding of the magnitude of the impact and strengthen the argument for addressing CC as a significant clinical concern;

-      Incorporating quotes or testimonials from patients and caregivers affected by CC could provide valuable insights into their experiences and the challenges they face. This would add a human element to the discussion and highlight the importance of addressing CC from a patient-centered perspective;

-  The decision to exclude evidence from pre-clinical models for the treatment of CC may limit the comprehensiveness of the review;

-      The section on pharmacologic interventions for cancer cachexia provides a comprehensive overview of current agents and their efficacy based on available evidence; however, the section could be further strengthened by providing a more balanced discussion of the potential risks and benefits of each pharmacologic intervention;

-     The section on new agents for managing cancer cachexia (CC) provides valuable insights into emerging pharmacologic interventions and their potential role in improving outcomes for patients; it could benefit from further elaboration on the potential mechanisms of action and specific patient populations that may benefit most from these interventions.

Congratulations to the Authors!

Author Response

We thank the reviewer for their valuable time, suggestions, and recommendations. We have added and edited our manuscript accordingly within the constraints of word limit set by the editors.

Abstract, “Could benefit from elaborating on specific outcomes of interventions mentioned, such as improvements in quality of life or survival rates, to provide a more comprehensive overview.” 

As per reviewers’ recommendation we have edited the abstract, indicating that Anamorelin improved appetite related QoL. We did not identify a study with an agent that showed improved survival rates.

“Providing a solid foundation for the reader to understand the scope and importance of the topic.”

As suggested by the reviewer, we have edited our introduction and added our objective for this review.

“The reference to the European Working Group on Sarcopenia in Older People (EWGSOP) consensus paper underscores the importance of integrating emerging research findings into clinical practice.”

We thank reviewer for their comment and valuable insight.

“Where possible, quantify the effects of CC on mortality, treatment response rates, quality of life scores, etc. This would provide a clearer understanding of the magnitude of the impact and strengthen the argument for addressing CC as a significant clinical concern.”

As per reviewers’ recommendation we have edited the clinical impact section and added relevant statistics, quantifications including mortality statistics.

“Incorporating quotes or testimonials from patients and caregivers affected by CC could provide valuable insights into their experiences and the challenges they face. This would add a human element to the discussion and highlight the importance of addressing CC from a patient-centered perspective.”

As suggested by the reviewer, in order to enhance the impact and share patient and caregivers’ perspective, we have included patient / caregiver perspectival quotes.

“The decision to exclude evidence from pre-clinical models for the treatment of CC may limit the comprehensiveness of the review.”

We thank reviewer for their comment and have included some pre-clinical studies where the evidence from clinical trials is especially limited (e.g., GDF-15 monoclonal antibody)

“The section on pharmacologic interventions for cancer cachexia provides a comprehensive overview of current agents and their efficacy based on available evidence; however, the section could be further strengthened by providing a more balanced discussion of the potential risks and benefits of each pharmacologic intervention.”

“The section on new agents for managing cancer cachexia (CC) provides valuable insights into emerging pharmacologic interventions and their potential role in improving outcomes for patients; it could benefit from further elaboration on the potential mechanisms of action and specific patient populations that may benefit most from these interventions.”

Where possible, we have briefly explained mechanism of action relating to CC and have referenced the relevant studies e.g. indicating that GDF-15, has potential for serving as both a marker and a therapeutic target in treating CC. Also stating that some agents e.g. corticosteroids are best reserved for specific patient populations (either patients with advanced cancer and/or exhibiting poor appetite and fatigue). Our figures also provide a framework for applying specific treatment options to the various mechanisms of cancer cachexia.  

Reviewer 2 Report

Comments and Suggestions for Authors

This is a well-written review for cancer cachexia (CC) focusing on clinical management and pharmacologic interventions for CC. As CC is currently hot topic for researchers and clinicians, this review provides useful information for clinicians who treat patients with CC. However, as CC is hot topic for researchers and clinicians, many research papers and review papers has been published. What is strong point of this review compared with previously published review papers? Please clearly indicate strong point of this review in introduction section. My impression is that the strong point of this review is updating information for clinical management and pharmacologic interventions for CC. Tille may be needed to revise to reflect main theme of this review. This review has introduced many pharmacologic agents for CC treatment and associated trials. If the table summarizing these matters are provided, it would be beneficial for readers to understand the contents.

Author Response

We thank the reviewer for their valuable time, suggestions, and recommendations. We have added and edited our manuscript accordingly within the constraints set by the editors.

Please clearly indicate strong point of this review in introduction section.

As suggested by the reviewer, we have edited introduction and added the objective of review.

Title may be needed to revise to reflect main theme of this review. 

We thank the reviewer for the recommendation, we have revised the title to: “Updates in Cancer Cachexia: Clinical management and Pharmacologic interventions”.

This review has introduced many pharmacologic agents for CC treatment and associated trials. If the table summarizing these matters are provided, it would be beneficial for readers to understand the contents.

We thank the reviewer for the helpful comment. We have tried our best to summarize potential management of CC in the graphical abstract and illustrated the selection of various pharmacologic agents based on the mechanism of cachexia in Figure 4. The most impactful strategies and pharmacologic agents for managing Nutritional impact symptoms are shown in the graphical abstract and Figure 4. We trust the figures showing specific pharmacologic agents will complement the text and allow readers to better understand the contents.

Reviewer 3 Report

Comments and Suggestions for Authors

 I read with real interest this review which discusses fundamental aspects of cancer cachexia.

The bibliography is updated and valid.

It is well written and well laid out.

Of note, the approach is critical and not just descriptive.

Author Response

We thank the reviewer for providing their valuable time to review our effort.

Round 2

Reviewer 1 Report

Comments and Suggestions for Authors

Congratulations to the authors for their work.